# A Preliminary Study on the Effect of Adding Sugarcane Syrup on the Flavor of Barley Lager Fermentation

**DOI:** 10.3390/foods13152339

**Published:** 2024-07-25

**Authors:** Hechao Lv, Yusheng Jia, Chaoyi Liu, Jia Xu, Caifeng Xie, Kai Li, Kai Huang, Fangxue Hang

**Affiliations:** 1College of Light Industry and Food Engineering, Guangxi University, Nanning 530004, China; lhcgzyx@126.com (H.L.); jys1562022@163.com (Y.J.); liuchaoyi32@163.com (C.L.); 15191008212@163.com (J.X.); fcx11@163.com (C.X.); gxlikai@gxu.edu.cn (K.L.); 2Provincial and Ministerial Collaborative Innovation Center for Sugar Industry, Nanning 530004, China; 3Engineering Research Center for Sugar Industry and Comprehensive Utilization, Ministry of Education, Nanning 530004, China; 4Guangxi Institute of Industrial Technology, Nanning 530001, China; kaige@gxgy.edu.cn

**Keywords:** sugarcane syrup, beer, process optimization, volatile substance, aroma stabilization

## Abstract

This study focuses on the diversified utilization of the sugarcane industry, and sugarcane syrup, as a by-product of the sugarcane industry, is a good raw material for fermentation. Bringing sugarcane syrup into beer is conducive to the enrichment of the sugar industry, and it can improve the flavor of beer and make it more aromatic. This study determined the optimal fermentation process for beer. By analyzing the consumption rate of the carbon and nitrogen sources of raw materials, the nutrient utilization of yeast, and the causes of differences in flavor substances, the flavor composition and flavor stability of beer were determined by SPME-HS-GC-MS technology. The results showed that beer brewed with sugarcane syrup as an auxiliary raw material met the basic specifications of beer. The addition of sugarcane syrup to the wort base increased the utilization of amino acids by the yeast, and LS (lager with added cane syrup) increased the nine flavor compounds of the beer, which constituted the basic flavor of the beer, bringing new flavor compounds compared with the normal all-barley beer. Forced aging experiments showed that LS produced fewer aging compounds than OWBL. Various experiments have shown that it is feasible to ferment beer with sugarcane syrup instead of partial wort.

## 1. Introduction

Lager beer is one of the world’s oldest and most commonly consumed alcoholic beverages. It is made from barley wort fermented by yeast at 10–15 °C [1]. The final aroma composition of the beer depends on the materials required for brewing and the parameters of the brewing process [2]. Volatile components in beer have a much greater impact on flavor profiles than in other alcohols [3]. Also, the detection of volatile substances is an important part of the process of the comprehensive evaluation of lager quality [4] and is the most intuitive way to analyze lager quality. Gas chromatography–mass spectrometry (GC-MS) coupled with headspace solid-phase microextraction (HS-SPME) has been widely used for the study of beer aroma [5], and electronic noses (e-noses) equipped with a series of electrochemical sensors have been shown to provide a comprehensive odor assessment [6]. Also, gas chromatography (GC) can be used to determine beer aging components. Studies have shown that carbonyl compounds are the main component of the “aging” taste of beer [7]. This means that measuring the amount of change in carbonyl compounds during the aging of beer is an important measure of flavor stability.

Unfavorable growing conditions for barley in some parts of the world (including those attributable to climate change) have led lager producers to use more versatile local raw materials to replace barley in conventional recipes. These materials are called adjuncts [8]. Syrup is one of the most commonly used liquid adjuncts. The use of syrup in lagers results in a lower production cost and a shorter fermentation cycle [9]. This fermentation has also been associated with altered (abnormal) patterns of sugar uptake and altered production of some flavor compounds [10].

China is one of the main sugarcane-producing countries, and Guangxi is the main sugarcane-growing area in China, accounting for more than 60% of China’s planting area. In the sugar production process, sugar cane is pressed into juice and then heated and concentrated to produce sugar cane syrup (sugar content 65–68°BX). Sugar cane syrup is rich in nutrients and is a quality brewing ingredient [11]. Sugarcane syrup is rich in amino acids, and its amino acid composition is different from that of wort. Changes in nutrient content can trigger different yeast reactions to produce beers with different flavors [2]. Sugar cane also contains many types of higher alcohols, which give beer a greater variety of flavor substances [6]. Moreover, studies have shown that sugarcane syrup contains a large number of polyphenolic components, including phenolic acids, flavonoids, and quinones [12], which have antioxidant properties. It is assumed that the polyphenolic components in sugarcane syrup can inhibit the oxidation of unsaturated fatty acids, isorhythmic ketones, higher alcohols, and other substances to form aldehydes and produce an “aging odor” [13]. Carbonyl compounds cause flavor changes in many foods, including milk, butter, vegetables, and oils. In 1966, Japanese researchers first found a significant increase in volatile carbonyl compounds in beer during storage and found that the increase in the aging flavor of stored beer coincided with an increase in the concentration of carbonyl compounds [14].

## 2. Materials and Methods

### 2.1. Experimental Material

*Saccharomyces cerevisiae var. diastaticus* with the trade name “BF 16” from Angie’s Yeast Company (China) was used for wort fermentation. The yeast was purchased as a solid fermenter, and the initial concentration of the medium after activation was 9 × 10^7^ cells/mL. The yeast liquid was added at 1% of the total wort volume according to the manufacturer’s recommendations. The yeast strain used was low flocculating and highly fermentable (76–80%) with an optimal fermentation temperature of 10–14 °C.

The materials for this study were three variants of Chinese lager: ordinary whole barley lager (OWBL), lager with added cane syrup (LS), commercially available lager 1 (CAL_1_), and commercially available lager 2 (CAL_2_). Ingredients include barley malt, wheat malt, light caramel malt, Munich malt, and red malt. All malts were sourced from Jinan Shuangmai Company (Jinan, Shandong, China). Citra (Qingdao, Shandong, China) and Cascade hops (Portland, OR, USA) were purchased from Guangzhou Dejussi Trading Company Limited (Guangzhou, China). Sugarcane syrup (Guangxi Baiguitang Co., Ltd., Chongzuo, Guangxi, China) was also added during the production process. The sugarcane syrup added during the production process was different from the traditional syrup made by chemical methods; after being pressed from sugarcane into sugarcane juice, it was pressed through a 50 nm membrane, which was evaporated and the filtrate was concentrated to 60–65°BX.

### 2.2. Physicochemical Parameters

The pH of finished beer was measured using a precision pH meter pH-3C. The extract content was measured at 20 °C using an Abbe refractometer, residual sugar was measured by DNS colorimetry [15], acidity was measured by sodium hydroxide titration, and the color of the beer was measured by a spectrophotometer at a wavelength of 420 nm. These measurements were repeated three times.

### 2.3. Determination of Bittering Value of Beer 

After the beer was degassed, 10 mL of degassed beer was acidified by adding 0.5 mL of 6 mol/L hydrochloric acid and then 20 mL of iso-octane, extracted by oscillation, centrifuged, and the absorbance was measured at 275 nm using a UV-visible spectrophotometer (Agilent 8453, Agilent, Santa Clara, CA, USA) [16].

The content of bitter substances in beer was calculated by the following formula.
X = A_275_ × 50
where X is the content of bitter substances in the sample, expressed in units of “BU”; A_275_ is the absorbance of the sample measured at 275 nm; and 50 is the conversion factor.

### 2.4. Beer Sensory Appetite Experiment

Twenty students (male:female = 1:1) who were physically fit and had a good sense of smell and taste were randomly selected to participate in the sensory evaluation according to the scoring rules table. The specific evaluation criteria can be found in the Appendix A.

### 2.5. Determination of Fermentable Sugars

A high-performance liquid chromatographic method was used for the quantification of fermentable sugars in wort and beer fermentation broth by an external standard [17]. Detection methods: chromatographic column: NH_2_ Analytical HPLC Column (4.6 × 250 mm, 5 μm); column temperature: 35 °C; differential refractive index detector; detector temperature: 35 °C; mobile phase: 75% acetonitrile; flow rate: 1.0 mL/min; injection volume: 20 μL. Standard preparation: quantitative glucose, fructose, maltose, sucrose, and maltotriose standards were dissolved in ultrapure water for gradient dilution and passed through a 0.22 μm membrane. Sample pretreatment: The wort and beer fermentation broth were centrifuged at 7000 rpm and then passed through a 0.22 μm membrane.

### 2.6. Determination of Free Amino Acids

The beer fermentation broth was analyzed using an automated amino acid analyzer and quantified using the external standard method [18]. Sample treatment: the samples were diluted 5 times with 4% sulfosalicylic acid solution, left at 20 °C for 30 min, centrifuged, and the supernatant was filtered with 0.22 μm nylon 66 membrane. 

### 2.7. Determination of Volatile Aroma Substances in Lager Beer

Electric nose: 1 mL of beer was diluted 10-fold and transferred into a sealed 100 mL conical flask and held at 25.0 °C ambient temperature for 60 min to allow the evaporation of volatile components. The volatile gas in the beaker was drawn into the sensor at a rate of 400 mL/min, and the computer recorded a time of 150 s. The sample was washed for 120 s before each measurement to remove the residual volatile gas and return the response value to the baseline state. The response value was selected from 145 to 150 s, and the average value was calculated. The measurement operation was repeated three times [19].

HS-SPME-GC-MS: HS-SPME conditions: a 5 mL glass vial containing 30 mL of beer sample was taken, 10 μL of 2-octanol at a concentration of 82.2 μg/mL was added as an internal standard, and the vial was closed with a cap with a PTFE cushion until analysis. 2-octanol was used as an internal standard, 2 g NaCl was added and rotated, and the vial was closed with a lid with a PTFE gasket until analysis. Equilibrium was achieved by pre-incubation for 10 min at 40 °C with magnetic stirring at 300 rpm. Experiments were conducted using a composite 50/30 m DVB/CAR/PDMS extraction head, and the head was aged for 20 min before each sample’s extraction to remove residues attached to the extraction head. The SPME fiber was extracted in the headspace of the sample for 40 min at 40 °C, then transferred to the injection port (230 °C) and left for 5 min for desorption before detection. The mixed standards were four alcohols: n-propanol, isoamyl alcohol, phenylethanol, and isobutanol. The standards were prepared as mixed standards and added to the internal standard injection, and their retention times and ion fragments were compared with those of the compounds in the samples. GC conditions: column: Agilent DB-Wax capillary column (60 m × 0.25 mm); carrier gas: helium, 99.999%; constant flow mode: 1.0 mL/min; etc. The temperature was initially 35 °C for 4 min, raised to 60 °C at 3 °C/min, held for 5 min, and finally raised to 230 °C for 15 min at 4 °C/min. The inlet temperature was 230 °C, and the sample was injected manually without splitting. The temperature was raised to 230 °C and finally held at 230 °C for 15 min. The inlet temperature was 230 °C, and the sample was injected manually without a split. MS conditions: ionization mode: EI; ionization energy: 70 eV; ion source temperature: 230 °C; quadrupole temperature: 150 °C; transfer line temperature: 270 °C; scan mode: SCAN; mass scan range: 32 to 300 amu; no solvent delay [13,20].

### 2.8. Determination of Antioxidant Power of Beer

Determination of antioxidant power of beer: Take 0.5 mL of beer sample diluted twice, add 2.5 mL of 6.5 × 10^−5^ mol/L DPPH ethanol solution, and react for 1 h at room temperature, then dilute the reaction solution 4-fold and move to a spectrophotometer for colorimetric comparison at a wavelength of 517 nm, and then compare its absorbance [21].

### 2.9. Determination of Volatile Aging Substances

In this experiment, OWBL and LS were used as research subjects, while LS and OWBL without sugarcane juice were used as control samples for the forced aging experiments. The aging aroma substances were stored at 45 °C and protected from light for a period of 9 d, and were characterized by GC-MS [22]. Standards: 2-furaldehyde, glutaraldehyde, phenylacetaldehyde, benzaldehyde, acetaldehyde, isobutyraldehyde, 2-methylbutyraldehyde, 3-methylbutyraldehyde, and trans-dinonenal. According to the chromatographic peaks, transfer 0.5 mL of each of the anti-dinonenal according to the different preparation methods and neutralize with ethanol to 200 mL to produce a mixed standard first-class reserve solution. Take 10 mL of 5% ethanol solution (pH 4.6–4.7 with 0.1% phosphoric acid) as the base, and add 1 μL, 2 μL, 4 μL, 8 μL, 15 μL, and 25 μL of the primary reserve solution to produce the standard sample solution. A 65 μm (PDMS-DVB) SPME fiber extraction head was used. HS-SPME conditions: a dilute solution of PFBOA was prepared by mixing 100 μL (6 g/L) of PBFOA solution with 10 mL of water in 30 mL glass vials and sealed with PTFE gasketed lids, and each vial containing diluted PFBOA was first equilibrated for 5 min at 50 °C and 250 rpm. The PDMS-DVB fibers were exposed to the headspace of the diluted PFBOA for 10 min. The SPME fibers with PFBOA were exposed to the headspace of the beer samples and extracted at 50 °C and 250 rpm for 45 min. The fibers were resolved in the GC inlet for 5 min [23]. GC conditions: gas chromatography was performed on an Agilent DB-Wax capillary column (60 m × 0.25 mm × 0.25 µm); carrier gas: helium, 99.999%; constant-flow mode: 1.0 mL/min; column warming program: initial oven temperature was maintained at 40 °C for 10 min, 10 °C/min up to 140 °C for 10 min, and 5 °C/min up to 250 °C for 10 min. The temperature of the inlet port was 250 °C, and the sample was injected manually without splitting [14]. MS conditions: ionization mode: EI; ionization energy: 70 eV, ion source temperature: 230 °C, quadrupole temperature: 150 °C; conversion line set at 270 °C; mass range: *m*/*z* 50–350; scanning mode: 4.65 scans/s. Carbonyl PFBOA derivatives were identified and the *m*/*z* 181 fragment was identified as the major fragment, and all aldehyde analyses were performed in the single-ion monitoring (SIM) mode, monitoring the ion *m*/*z* 181. Beer was also analyzed by gas chromatography/mass spectrometry (GC/MS) without PFBOA derivatization [24].

### 2.10. Methods of Data Analysis

Data were processed and plotted using Excel, SPSS 24.0, Xcalibur 4.0, and Origin 9.0.

Experimental GC-MS data of the samples were processed using Xcalibur 4.0 software and compared by computerized search and spectral library. The mixed C8–C40 standards were analyzed under the same GC-MS conditions and the retention indices (RIs) of the components to be measured were calculated according to the following equation.

Retention index
RI = 100z + 10 [TR(x) − TR(z)] TR(z + 1) − TR] RI = 100z + 10 [TR(x) − TR(z)] TR(z + 1) − TR

TR(x): retention time of the component to be measured;

TR(z): Retention time of n-alkanes with carbon atom number z;

TR(z + 1): Retention time of n-alkanes with z + 1 carbon atoms.

## 3. Results and Discussion

### 3.1. Influence of Process Parameters on the Finished Beer

In order to discuss the effects of wort concentration and sucrose syrup addition ratio on the indexes during beer fermentation, a brewing one-factor optimization experiment was carried out to ensure that other conditions were the same.

The above Table 1 compares all LSs brewed at different concentrations. The sugar cane juice and the wort itself were close in color, both being clear amber–yellow, and there was little difference in the appearance of gloss and color when comparing the different percentages of sugar juice added during the pre-fermentation period. The raw wort concentration had a significant effect on the alcohol, total acid, residual sugar, color, and bitterness values of the beers, which derived their bitterness mainly from hop resins and, to a much lesser extent, from non-hop components, such as bittering peptides, amino acids, and polyphenols produced during the brewing process [25]. Saccharomyces cerevisiae takes up the amino acids present in the wort, which they bring with them from the wort with an amino group so that they can be integrated into its own structure. The remaining amino acids (α-keto acids) enter into an irreversible chain reaction that culminates in the formation of the byproducts, higher alcohols [3]. A moderate amount of higher alcohol in beer will give it a rich, full-bodied feel, but if the level is too high, it will give the beer a serious “after-bitter” taste and cause the drinker to go “on the head”. Esters are the most important aroma substances produced by yeast. They have a very low odor threshold in beer and greatly determine the final aroma [26]. Beer produces organic acids under aerobic conditions, a pathway thought to be a byproduct of the tricarboxylic acid cycle [26]. As the amount of organic acids produced by this process increases, the taste of the beer develops an unpleasant “sourness” that degrades the quality of the beer [27].

As can be seen from the Table 2, the beer with 25% addition had the highest alcohol content of 7.2%, and the sugar cane beer with 30% and 50% addition had alcohol contents of 6.6% to 6.7%, with higher residual sugar contents of 9.25 g/L and 12.50 g/L, respectively. The yeast had limited metabolic capacity in the post-fermentation period (uncoordinated carbon-to-nitrogen ratio) and could not fully utilize the supplemented carbon source, resulting in a high residual sugar content in the beer. Experiments have shown that the addition of juice in the range of less than 25% has good fermentability and low residual sugar. The total acidity does not correspond quantitatively with the pH value because the beer contains a lot of H^+^ in a non-free state. With a sugar cane juice pH of 5.13 and the addition of 5% to 25% juice, the total acidity of the beer ranged from 1.53 to 1.67 mL/100 mL, which was lower than the 1.96 mL/100 mL of OWBL, and exceeding 25% juice addition increase the acidity of the beer. From the table, it seems that the amount of juice addition was not linearly related to the bitterness value and color of the beer, which were related to the fermentation metabolic state of the brewer’s yeast. After the sensory panel tasting, the fruit aroma of the added sugar cane juice increased with the addition ratio; as the fruit aroma increased, the sweetness of the beer became stronger, and the corresponding hop flavor became lighter. Sugarcane syrup addition and advanced alcohol content were nonlinearly correlated. We were unable to determine the relationship between the amount of syrup added and the content of higher alcohols. When 20% sugar cane juice was added, the fruit aroma and hop aroma were more harmonious, and the taste was suitable. Comparing the different concentrations of LSs in the table, the raw wort concentration had a significant effect on the alcoholic strength, total acid, residual sugar, color value, and bitterness value of the beers. The bitterness value decreased with increasing concentration because the wort concentration affected the leaching of bitter substances from hops during boiling. As the wort concentration increased, the ethanol content of the brew increased, and the higher alcohol content also increased. Measuring the sensory data and the advanced alcohol content, the preference was higher for 10°P pale lager.

The fermentation cycle of pale lagers fermented at different temperatures varied, As expressed in Table 3. With the shortest fermentation time at 14 °C requiring only 6 d for primary fermentation (high-foam phase), while 8 °C and 12 °C required 8–12 d for primary fermentation. The higher the temperature, the better the utilization of the carbon source, and the lowest residual sugar was obtained, with a residual sugar content of 8.06 g/L for the beer at 14 °C and 11.28 g/L for the beer at 8 °C. The temperature had an effect on the production of higher alcohols in the beer. The fermentation at 8 °C was slow, and the highest residual sugar content was 11.28 g/L. Temperature had an effect on the production of higher alcohols in beer, and from the table, it can be seen that higher temperatures produced higher levels of higher alcohols than OWBLs. At the same time, the concentration of higher alcohols also increases, and too many higher alcohols reduce the quality of beer. Similarly, high fermentation temperatures can lead to the production of organic acids, which can reduce the quality of beer.

### 3.2. Determination of the Optimal Fermentation Process

The addition of hops can bring bitterness to the beer and enrich its flavor hierarchy. The bitterness of beer mainly comes from hop resins, and a small portion of it comes from non-hop components, such as bittering peptides, amino acids, polyphenols, and so on, during the brewing process, among which iso-alpha-acids contribute the most to the bitterness of beer [28]. And the addition of hops can clarify the wort by complexing and precipitating the proteins in the wort during the boiling process. Various concentrations of hops were added to each brew, and the sensory evaluation results showed that the bittering value of the beer was most acceptable in the range of 13–15, which corresponded to hops addition of 0.04%.

The eligible samples were selected for response orthogonal experiments, which required the basic indexes of brewing beer: the advanced alcohol value was between 50 and 120 mg/L, the normal value of diacetyl was between 0 mg/L and 0.1 mg/L, the acidity was ≤2.6 mg/100 mL, the mass fraction of carbon dioxide was above 3.5, and the alcohol content was ≥3.7%. In the sensory evaluation experiments, we found higher sensory evaluation scores for beers that better met the beer rating criteria, so we chose the total sensory tasting score as the only criterion for evaluating the quality of beer for the response surface experiments, and we conducted the response surface experiments with the wort concentration, the amount of syrup added, amount of hops added, and the fermentation temperature as the factors for the experimental experiments. (The sensory evaluation criteria can be found in the Appendix A). The results of the response orthogonal experiment are as Table 4:

According to the R values in Table 5, the main and secondary influences of individual factors on sensory evaluation are: D > A > B > C, i.e., hop addition > wine addition > wine concentration. That is to say, the amount of hop addition > the amount of juice addition > the concentration of original wort > the fermentation temperature. According to the K of each factor, the optimized condition is A_2_B_2_C_2_D_2_. According to the experimental verification, it was unanimously evaluated that 10°P LS had better color and taste, which meant that A_2_B_2_C_2_D_2_ was the optimal condition, which is in line with the results of the one-way experiment. A_2_B_2_C_2_D_2_ was concluded to be the optimal fermentation condition. The final results were: 10°P wort concentration as the base, addition of 20% sugarcane syrup, fermentation temperature of 12 °C, and an ale hops addition of 0.04%.

The basic physicochemical indexes of sugarcane lager beer and OWBL under the same conditions are listed in the following Table 6.

### 3.3. Changes in Substances during Fermentation

As the same fermentation conditions were used (different fermentation substrates (carbon and nitrogen sources)), the resulting beers had different basic indicators. The carbon and nitrogen sources directly influence the rate of yeast fermentation and the expression of the entire flavor profile of beer (alcohols, esters, aldehydes, and ketones), which in turn affect the overall flavor profile of beer [1]. The analysis of the fermentation substrate depletion in lagers is an essential step in judging the quality of lager beers. Two main changes occur during the fermentation of lager: the conversion of nitrogenous compounds and the fermentation of sugars. Other by-products are also produced along with these two changes.

#### 3.3.1. Fermentable Sugars

Comparing the sugar consumption during the fermentation of normal beer and LS, the results of the sugar spectrum analysis for 10°P wort were 2.43 g/L for fructose, 8.39 g/L for glucose, 1.42 g/L for sucrose, 80.60 g/L for maltose, and 5.28 g/L for maltotriose. High-performance liquid-phase sugar analysis for 10°P sugarcane syrup juice raw material showed that the fructose content was 0.79 g/L, the glucose content was 0.63 g/L, and the sucrose content was 11.58 g/L. The main sugar in the sugarcane syrup was sucrose.

As shown in Table 7, yeast itself will preferentially utilize sucrose [29], which breaks down the non-reducing sugar (sucrose) in wort into one molecule of fructose and one molecule of glucose under the action of sucrose converting enzyme. The addition of sucrose from sucrose juice increases the amount of glucose, and glucose deterrence prevents the expression of the MAL gene (which affects the uptake of maltose and maltotriose) when the glucose concentration exceeds 1% and allows the yeast to fully utilize the glucose, with a small amount of fructose remaining at the end of fermentation; fructose is twice as sweet as sucrose per mole, and the higher the fructose residue, the sweeter the final beer product; 10°P LS has a lower maltose content than OWBL, and the addition of cane syrup results in a change in the ratio of sugars that causes the residual fructose and sucrose to rise.

#### 3.3.2. Free Amino Nitrogen

Wort contains a large amount of nitrogenous substances, but not all of these substances are utilized by yeast in cellular metabolic pathways. Nitrogen-containing compounds in wort are mainly composed of free amino nitrogen, ammonia ions, and small molecular peptides (dipeptides and tripeptides), and yeast produces different volatile aromas due to the utilization of different ratios of amino acids, i.e., amino nitrogen is the main source of variations in aroma substances in beer.

The Table 8 shows the amino acid contents in LS, OWBL, wort, and sugarcane syrup. It can be inferred that LS has a higher and more comprehensive utilization of amino nitrogen as compared with OWBL. The amino nitrogen content of LS was significantly lower than that of normal whole-barley beer. It is hypothesized that this is due to the addition of sugarcane syrup at the fermentation stage altering the yeast metabolic pathway, allowing the yeast to newly utilize amino acids (e.g., histidine) that are not readily available in OWBL. LS utilizes amino acids at a higher rate than OWBL. Fewer amino acids remain, which makes LS less susceptible to contamination by spoilage bacteria that utilize amino acids for growth, such as lactobacilli and lactococci. Fruit juices contain 19 of the 20 essential amino acids (without cysteine), and just as wort utilizes sugar, the amino acids are taken up by yeast cells in the order in which they are taken up [30]. The amino acids of yeast are classified into two groups (a, b) based on the percentage of amino acids utilized by the yeast; the a group of amino acids is more readily utilized by the yeast and utilized in a higher percentage, and the b group of amino acids is utilized in a smaller percentage. It can be said that the addition of sugarcane syrup can promote the utilization of b-amino acids by yeast and change the fermentation pattern of yeast, thus producing different flavor substances.

### 3.4. Detection of Aroma Substances in Beer

#### 3.4.1. The Electronic Nose

Principal component analysis (PCA) and latent Dirichlet allocation (LDA) were chosen for the analysis of e-nose data, where beer has a wide variety of volatiles, and PCA was used to analyze different broad categories of flavor substances in lagers, while LDA was used to determine the variability of the broad categories of substances among beers.

From the figure, it can be seen that the contribution rates of the first and second principal components were 73.51% and 20.49%, respectively, and the total contribution rate was 94%, which can represent the characteristic information of the broad categories of volatiles of the samples. The flavor differences between the laboratory lager and the commercially available lagers were evident in PC_2_, while PC_1_ was close to that of the commercially available lager. The difference between PC_1_ and PC_2_ of the sugarcane lager and those of the ordinary and commercially available lagers was significant. This proves that the aroma of sugarcane lager is different from that of commercially available lager.

The aroma of sugarcane lager is different from that of commercially available lager and laboratory lager, which is mainly due to the different raw materials and fermentation processes, as represented in Figure 1. The direction chosen by PCA maintains maximum structure between data in lower dimensions, while the direction chosen by LDA achieves maximum separation between the given classes [30]. The LDA classification results are more representative of the variability than the PCA results. The data collection points of the same type of lager under the same conditions in the ellipse in Figure 1b represent the fingerprint profile of that product type, and the more dense data points represent the higher repetition rate of that sample. As can be seen from the figure, the contribution of LDA discriminant LD_1_ and LD_2_ for lager samples were 98.64% and 0.93%, respectively, and the total sum was 99.57%, which can represent all the characteristic information of the samples. The data collected from each type of lager samples were distributed in different areas without overlapping and far away from each other; this indicates that there were significant differences in odor. This indicates that there were differences in the flavor quality of lagers brewed from different raw materials.

As can be seen from the Figure 2, all the lighter samples had the highest response values for the three sensor types W5S, W1S, W1W, and W2W. The high response value for ammonia–oxygen compounds is due to the high sensitivity of the detector itself. The substances corresponding to inorganic sulfides in lager are sulfides, such as SO_2_, which is a metabolic product of yeast in the fermentation process. The amount of SO_2_ produced is related to the yeast strain, wort concentration, and fermentation process [31]. The high response value of the corresponding methyl group sensor in the figure is mainly due to the high content of 2-methyl sulfide substances in lager. In nature, 2-methyl sulfide is often produced by the decomposition of proteins and has a “green” and “fruity” aroma, i.e., sugarcane syrup can provide lager a high “green” and “fruity” aroma, and “fruity” aroma.

#### 3.4.2. GC-MS

There were a total of 77 compounds in the GC-MS assay, and the beer compounds detected by GC-MS were divided into three categories: compounds with no significant difference, compounds with more LS than OWBL, and compounds with less LS than OWBL, which were added to the commercially available beer assay to test for volatiles of roughly the same composition as the commercially available beer. The reason for adding the commercially available beer was to test the volatiles to ensure that the brewing process was approximately the same as the commercially available pale lager. In this experiment, substances specific to a single commercially available lager were ignored, thus minimizing the effects of the use of different hop types and different fermentation processes, as well as different flavor substances produced by different brewing yeasts. The results are shown in Table 9.

It is the aroma profile of the LS that is illustrated in Table 9. The combination of GC-MS and an electronic nose can only characterize the composition of the flavor substances in beer and determine its flavor components. The characteristic aroma varies from lager to lager, and this aroma comes partly from the aroma of the raw materials and adjuncts themselves and partly from the yeast or metabolites produced during their fermentation. There are a total of 77 compounds in Table 9, and 51 substances were detected in sugarcane lager. Fifty-two compounds were found to be the same in the two commercially available lagers compared, of which esters and alcohols accounted for 80%, indicating that in sugarcane lager, alcohols and esters constitute the basic flavor; thirty-two compounds were common to the four lagers, and eleven substances were unique to sugarcane lager compared with the other three lagers, namely camphor, carbonochl, ethyl(E) cinnamate, acetophenone, 4-ethylbenzaldehyde, cyclododecane, ethyl myristate, m-phthalaldehyde, phenethyl camphor, carbonochl, and ethyl(E) cinnamate, which were found to be contained in the sugarcane syrup itself after comparison with the sugarcane syrup and wort components. These are the flavor substances of the sugarcane syrup itself and the flavor substances obtained by yeast fermentation of the sugarcane syrup. At the same time, compared with commercially available whole-barley lagers, laboratory whole-barley lagers produce unique substances, such as 2-acetylpyrrole, gamma-nonanolactone, etc. These substances can be considered to be produced by the overfermentation of wort by yeast under laboratory conditions, but they are not present in laboratory sugarcane lagers, and the possible reason for this analysis is because brewer’s yeast can produce substances such as acetophenone by the joint action of wort and sugarcane syrup, but it is not possible to conclude that acetophenone, 4-ethylbenzaldehyde, cyclododecane, ethyl myristate, m-phthalaldehyde, phenethyl hexanoate, methanone, (4-ethylphenyl)phenyl-, and ethyl palmitate are produced by the yeast metabolism of sugarcane syrup or by yeast synergism of sugarcane syrup and wort. It is presumed that it is mainly produced by the yeast using amino acids (such as histidine) that are not readily available in the post-fermentation stage, combined with the sugar in the sugar cane syrup.

### 3.5. Determination of Lager Aging Characteristic Aroma

The comparison of aldehydes in lagers before and after aging is shown in Table 10: 2-furaldehyde, acetaldehyde, isobutyraldehyde, glutaraldehyde, phenylacetaldehyde, and 3-methylbutyraldehyde. 2-Furaldehyde was the main heat load indicator and was not detected in commercially available lager, sugar cane lager, or OWBL before aging, while 2-furaldehyde increased to varying degrees in all lagers after aging, which can be attributed to residual carbohydrate reactions, with sugar cane lager having significantly lower residual sugar than OWBL, which had the lowest production of furfuraldehyde. From the data in the table, it can be seen that the degree of accumulation of 2-furaldehyde in sugarcane lager was lower than those of OWBL and commercially available lager, i.e., sugarcane lager had a higher level of resistance to aging than ordinary lager, which contributed to its flavor stability. However, in an experiment to measure the antioxidant power of beer, we found that the antioxidant power of LS was about 10% lower than that of OWBL. It is hypothesized that this was due to the fact that the free amino acid residue at the completion of fermentation of LS was smaller than that of OWBL, resulting in a weaker oxidation of amino acids. The increase in phenylacetaldehyde in sugarcane lager from 1.861 µg/L to 10.744 µg/L was higher due to Strecker degradation of amino acids to form aldehydes, such as 2-methylpropionaldehyde (isobutyraldehyde) and phenylacetaldehyde [32]. But, whether it is the effect of sugarcane syrup remains to be studied. The increase in glutaraldehyde during lager aging is then due to the autoxidation of linoleic acid. During forced aging, the fruity aroma of the lager gradually fades at first and the hops flavor gradually disappears, followed by a strong “currant flavor” and “soy sauce flavor”, and the longer the aging time, the heavier the “currant flavor”. Saison [33] described the aging flavors after sensory panel evaluation as “cardboard taste”, “metallic taste”, solvent taste”, “old hops taste”, “cool chestnut taste”, “merlot taste”, “thioether taste ”, “acetaldehyde” (“green apple”), and “white wine”. The main substance of “cardboard flavor” is trans-2-nonenal (T2N) [23], but it was not detected in the present study. Some researchers have suggested that aging flavors differ by lager type [34], and the “cardboard taste” in lagers is not the only characteristic aging taste of lager.

## 4. Conclusions

The present study was a preliminary investigation of the effect of brewing lager beer with sugarcane syrup instead of some malt on the quality of the resulting beer. The optimal fermentation process for sugarcane beer was confirmed by one-way and orthogonal experiments, and it was found that the effect of syrup addition on the organoleptic flavor of the beer was smaller than that of hops addition, and larger than that of fermentation temperature and wort concentration. Analysis of the fermentation process showed that the beer with partial replacement of wort with sugarcane syrup had a higher utilization of free amino nitrogen and more complete fermentation. The experiments showed that sugarcane lager flavor substances constituted the basic flavor of lager beer, while because of the different ratio of amino acids, some flavor substances in sugarcane lager beer had differentiation from OWBL.

Volatile aldehydes are the most intuitive indicator for evaluating the quality and degree of aging of lagers, but judging the quality and aging rate of lagers by volatile matter is incomplete. Although lagers with sucrose syrup as an adjunct in volatile content detection have unique flavor substances and higher flavor stability compared with whole-barley lagers, the specific mechanism of sucrose syrup’s effect on the fermentation of lagers is not yet clear, and the research direction brought by sucrose syrup as an adjunct to lager beer is still extensive. In addition to the detection of volatile substances, it is necessary to determine the mechanism by which sugarcane syrup promotes yeast utilization of amino acids that are not readily available and to comprehensively evaluate the quality of lagers with sugarcane syrup as an adjunct so that sugarcane, a saccharide, can be used in a more diverse range of products.

## Figures and Tables

**Figure 1 foods-13-02339-f001:**
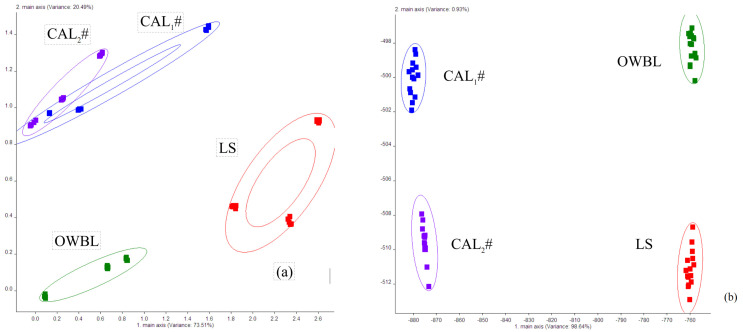
Electronic nose fingerprints of laboratory lager and market lager. (**a**) Principal component analysis results; (**b**) LDA analysis results.

**Figure 2 foods-13-02339-f002:**
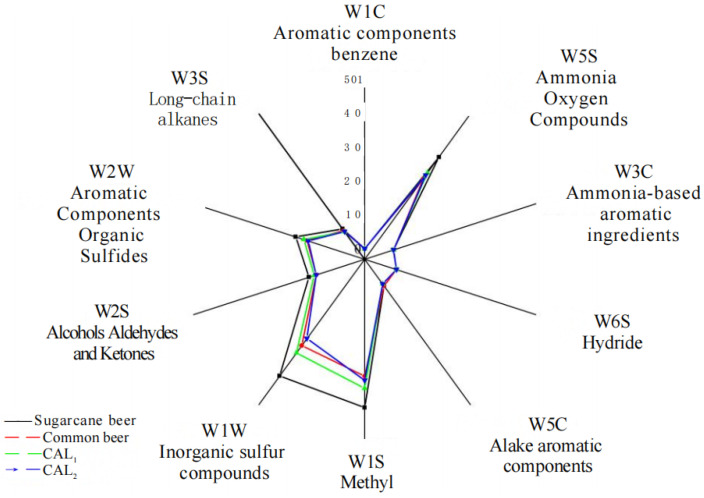
Electronic nose radar map of home-brewed lager in the laboratory and lager on the market.

**Table 1 foods-13-02339-t001:** Effect of different sugarcane juice additions on fermentation of sugarcane beer (original wort of the beer is 12°P, fermentation at 12 °C).

Addition of Sugar Cane Syrup	Physical and Chemical Indicators Related to Sugar Cane Beer						
Alcoholic Strength (%)	Total Acidity (mL/100 mL)	pH	Residual Sugar (g/L)	Color (EBC)	Bitterness Value (BU)	Senior Alcohol (mg/L)
OWBL (10°P)	5.6 ^d^ ± 0.3	1.96 ^c^ ± 0.10	4.77 ^c^ ± 0.02	9.41 ^b^ ± 0.02	13.35 ^bc^ ± 0.32	19.8 ^a^ ± 1.0	116.67 ^a^ ± 22.53
5%	6.0 ^c^ ± 0.1	1.53 ^f^ ± 0.11	4.78 ^bc^ ± 0.01	9.55 ^b^ ± 0.04	12.88 ^c^ ± 0.36	18.0 ^b^ ± 0.5	94.39 ^b^ ± 18.64
10%	6.5 ^b^ ± 0.1	1.53 ^f^ ± 0.11	4.77 ^bc^ ± 0.00	9.19 ^b^ ± 0.13	13.61 ^b^ ± 0.16	17.4 ^b^ ± 1.4	97.00 ^b^ ± 20.01
15%	6.1 ^c^ ± 0.2	1.57 ^ef^ ± 0.03	4.78 ^bc^ ± 0.01	9.40 ^b^ ± 0.21	12.78 ^c^ ± 0.20	16.6 ^bc^ ± 0.9	100.94 ^bc^ ± 19.05
20%	6.2 ^c^ ± 0.1	1.63 ^de^ ± 0.01	4.80 ^b^ ± 0.01	7.65 ^c^ ± 1.23	13.05 ^bc^ ± 0.67	16.3 ^bc^ ± 0.5	100.83 ^bc^ ± 24.56
25%	7.2 ^a^ ± 0.1	1.67 ^d^ ± 0.03	4.61 ^d^ ± 0.00	8.72 ^b^ ± 0.04	13.24 ^bc^ ± 0.24	16.6 ^bc^ ± 1.2	119.72 ^a^ ± 16.88
30%	6.6 ^b^ ± 0.1	2.30 ^a^ ± 0.01	4.63 ^d^ ± 0.01	9.25 ^b^ ± 0.51	13.04 ^bc^ ± 0.21	15.4 ^c^ ± 0.6	108.45 ^c^ ± 24.23
50%	6.7 ^b^ ± 0.1	2.20 ^b^ ± 0.00	4.83 ^a^ ± 0.01	12.50 ^a^ ± 0.51	17.6 ^a^ ± 0.20	13.6 ^d^ ± 0.9	105.88 ^c^ ± 19.65

(Note: The data in this table are expressed as the mean (*n* = 3) ± standard deviation, using Duncan’s test, and the difference is significant if the superscript letters are different (*p* < 0.05)).

**Table 2 foods-13-02339-t002:** Effect of different wort concentrations on fermentation of sugarcane beer (beer with 20% syrup addition, fermentation at 12 °C).

Wort Consistency	Physical and Chemical Indicators Related to Sugar Cane Beer						
Alcoholic Strength (%)	Total Acidity (mL/100 mL)	pH	Residual Sugar (g/L)	Color (EBC)	Bitterness Value (BU)	Senior Alcohol (mg/L)
6°P	3.6 ^e^ ± 0.1	1.21 ^c^ ± 0.10	4.48 ^b^ ± 0.13	5.60 ^e^ ± 0.41	7.54 ^c^ ± 0.59	17.1 ^a^ ± 0.1	92.32 ^a^ ± 17.75
8°P	4.4 ^d^ ± 0.0	1.23 ^c^ ± 0.03	4.54 ^b^ ± 0.04	7.52 ^d^ ± 0.23	10.40 ^bc^ ± 1.13	15.6 ^b^ ± 0.0	97.67 ^a^ ± 20.54
10°P	5.2 ^c^ ± 0.0	1.49 ^bc^ ± 0.41	4.62 ^ab^ ± 0.10	10.13 ^c^ ± 0.43	12.42 ^b^ ± 0.66	14.2 ^c^ ± 0.1	103.48 ^ab^ ± 25.12
12°P	6.2 ^b^ ± 0.0	1.88 ^ab^ ± 0.12	4.73 ^a^ ± 0.01	12.54 ^b^ ± 0.30	13.65 ^b^ ± 0.42	11.9 ^d^ ± 1.1	106.35 ^ab^ ± 18.04
14°P	7.2 ^a^ ± 0.0	2.19 ^a^ ± 0.24	4.63 ^ab^ ± 0.10	15.41 ^a^ ± 0.69	18.58 ^a^ ± 4.48	9.2 ^e^ ± 0.5	119.2 ^b^ ± 18.91
OWBL (10°P)	5.6 ^d^ ± 0.3	1.96 ^c^ ± 0.10	4.77 ^c^ ± 0.02	9.41 ^b^ ± 0.02	13.35 ^bc^ ± 0.32	19.8 ^a^ ± 1.0	114.25 ^b^ ± 22.06

(Note: The data in this table are expressed as the mean (*n* = 3) ± standard deviation, using Duncan’s test, and the difference is significant if the superscript letters are different (*p* < 0.05)).

**Table 3 foods-13-02339-t003:** Effect of different fermentation temperatures on fermentation of sugarcane beer (beer had a wort strength of 10°P and syrup addition of 20%).

Temp	Physical and Chemical Indicators Related to Sugar Cane Beer						
Alcoholic Strength (%)	Total Acidity (mL/100 mL)	pH	Residual Sugar (g/L)	Color (EBC)	Bitterness Value (BU)	Higher Alcohol (mg/L)
10 °C	5.3 ^b^ ± 0.0	1.60 ^b^ ± 0.01	4.82 ^a^ ± 0.01	11.28 ^a^ ± 0.51	14.3 ^b^ ± 0.30	20.5 ^a^ ± 1.0	90.24 ^a^ ± 19.23
12 °C	5.3 ^b^ ± 0.1	1.60 ^b^ ± 0.01	4.68 ^a^ ± 0.01	10.13 ^b^ ± 0.25	14.7 ^ab^ ± 0.30	17.3 ^b^ ± 1.5	109.15 ^b^ ± 18.69
14 °C	5.7 ^a^ ± 0.0	1.80 ^a^ ± 0.10	4.86 ^a^ ± 0.01	8.06 ^c^ ± 0.31	15.20 ^a^ ± 0.10	19.05 ^ab^ ± 1.1	124.49 ^c^ ± 22.72

(Note: The data in this table are expressed as the mean (*n* = 3) ± standard deviation, using Duncan’s test, and the difference is significant if the superscript letters are different (*p* < 0.05)).

**Table 4 foods-13-02339-t004:** Orthogonal experimental factors and levels.

Level	Factors			
Amount of Syrup Added (%)	Wort Consistency (°P)	Fermentation Temperature (°C)	Hops Added (%)
1	10	8	8	0.02
2	20	10	10	0.04
3	30	12	12	0.06

**Table 5 foods-13-02339-t005:** Orthogonal experiment results.

Experiment Number	A	B	C	D	Sensory Evaluation
1	1	1	1	1	76
2	1	2	2	2	91
3	1	3	3	3	69
4	2	1	2	3	72
5	2	2	3	1	87
6	2	3	1	2	86
7	3	1	3	2	78
8	3	2	1	3	62
9	3	3	2	1	73
K_1_	236	226	224	236	
K_2_	245	240	236	255	
K_3_	213	228	234	203	
k_1_	78.667	75.333	74.667	78.667	
k_2_	81.667	80.000	78.667	85.000	
k_3_	71.000	76.000	78.000	67.667	
R	10.667	4.667	4.000	17.333	

**Table 6 foods-13-02339-t006:** Final physical and chemical indicators for Ls and OWBL.

Project	Standards	LS	OWBL (10°P)
Alcoholic strength (%)	10.1°P–11.0°P ≥ 3.7	4.9–5.3	4.7–4.9
Total acidity (mg/100 mL)	10.1°P–14.0°P ≤ 2.6	1.0–2.0	1.6–2.4
Diacetyl (mg/L)	≤0.10	0.07	0.09
CO_2_ (mass fraction)	0.35–0.65	0.40	0.45
Higher alcohol (mg/L)	50–120	109.15	105.42
Amount of hops added (%)	Bitterness value 13 ≤ x ≤ 15	14.01	14.9

**Table 7 foods-13-02339-t007:** Measurement of the amount of fermentable sugars before and after fermentation.

	Glucose (g/L)	Fructose (g/L)	Maltose (g/L)	Sucrose (g/L)	Maltotriose (g/L)
10°P Wort	8.39 ± 0.26	2.43 ± 0.11	80.6 ± 6.8	1.42 ± 0.02	5.28 ± 0.42
10°P Cane Juice	0.63 ± 0.03	0.79 ± 0.05	0	11.58 ± 1.35	0
10°P OWBL (mg/100 mL)	0.68 ± 0.02	0.12 ± 0.01	3.07 ± 0.22	0.87 ± 0.03	0.29 ± 0.01
LS (10°P substrate) (mg/100 mL)	1.22 ± 0.03	0.36 ± 0.01	2.62 ± 0.12	0.92 ± 0.12	0.38 ± 0.01

**Table 8 foods-13-02339-t008:** Amount of change in free amino acids and available nitrogen sources before and after fermentation.

Amino Acid Type	LS (10°P Substrate) (mg/100 mL)	10°P OWBL (mg/100 mL)	Control Wort (mg/100 mL)	Sugar Cane Juice (mg/100 mL)
Asp	5.85 ± 0.47	7.22 ± 0.64	9.17 ± 0.78	13.45 ± 1.04
Thr	0.93 ± 0.04	2.13 ± 0.18	6.79 ± 0.47	2.92 ± 0.34
Ser	1.91 ± 0.16	2.35 ± 0.28	6.88 ± 5.35	8.41 ± 0.96
Glu	14.29 ± 1.54	8.03 ± 0.78	13.24 ± 1.64	81.56 ± 8.13
Gly	2.96 ± 0.13	4.16 ± 0.31	3.88 ± 0.28	0.79 ± 0.04
Ala	8.99 ± 0.68	12.2 ± 1.51	11.2 ± 1.67	7.35 ± 0.76
Cys	0.24 ± 0.03	0.37 ± 0.02	0.54 ± 0.06	0.11 ± 0.01
Val	2.54 ± 0.03	8.72 ± 0.79	12.32 ± 1.36	5.25 ± 0.04
Met	0.47 ± 0.04	2.12 ± 0.31	4.33 ± 0.42	0.64 ± 0.05
Ile	0.76 ± 0.06	4.02 ± 0.36	7.54 ± 0.69	2.83 ± 0.29
Leu	0.78 ± 0.08	7.12 ± 0.82	16.60 ± 1.72	2.16 ± 0.23
Tyr	6.06 ± 0.53	10.78 ± 1.32	12.96 ± 1.32	2.26 ± 0.19
Phe	2.77 ± 0.31	11.26 ± 1.41	16.33 ± 1.75	2.12 ± 0.18
Lys	0 ± 0	5.56 ± 0.53	10.07 ± 0.99	0.79 ± 0.09
NH3	1.46 ± 0.14	2.56 ± 0.28	3.12 ± 0.35	0.68 ± 0.07
His	1.49 ± 0.11	4.15 ± 0.40	5.61 ± 0.61	2.67 ± 0.19
Arg	1.19 ± 0.09	8.49 ± 0.72	11.7 ± 1.25	1.34 ± 0.13
Hyp	1.53 ± 0.08	2.81 ± 0.19	7.94 ± 0.81	3.19 ± 0.27
Pro	35.81 ± 2.67	41.71 ± 5.96	41.98 ± 3.78	3.16 ± 0.21

**Table 9 foods-13-02339-t009:** Volatile components of homemade LS, OWBL, and market beer.

NO	Volatile Compounds	RI	CAS	Relative Content (%)				Aroma Characteristic
				OWBL	LS	CAL_1_	CAL_2_	
1	Acetaldehyde	710	75-07-0	0.082 ^a^ ± 0.014	NQ	0.031 ^b^ ± 0.006	NQ	Pungent, ethereal, aldehydic, fruity
2	Isobutyl acetate	997	110-19-0	0.038 ^a^ ± 0.007	NQ	0.068 ^b^ ± 0.014	0.141 ^c^ ± 0.020	Sweet, fruity, ethereal, banana, tropical
3	Ethyl valerate	1118	539-82-2	0.028 ^a^ ± 0.004	NQ	NQ	NQ	Sweet, fruity, apple, pineapple, green, tropical
4	2-Heptanone	1126	110-43-0	0.011 ^a^ ± 0.001	NQ	NQ	NQ	Fruity, spicy, sweet, herbal, coconut, woody
5	Dipentene	1140	7705-14-8	0.046 ^a^ ± 0.007	NQ	0.033 ^b^ ± 0.009	NQ	Citrus, herbal, terpene, camphor
6	Methylheptenone	1295	110-93-0	0.024 ^a^ ± 0.004	NQ	NQ	0.024 ^a^ ± 0.005	Citrus, green, musty Lemongrass, apple
7	2-Nonanone	1385	821-55-6	0.045 ^a^ ± 0.007	0.021 ^b^ ± 0.004	0.026 ^b^ ± 0.003	0.026 ^b^ ± 0.003	Fresh, sweet, green, weedy, earthy, herbal
8	Decanal	1545	112-31-2	1.152 ^a^ ± 0.326	0.141 ^b^ ± 0.003	0.187 ^b^ ± 0.002	0.511 ^c^ ± 0.011	Sweet, aldehydic, waxy, orange peel, citrus, floral
9	Octanol	1597	111-87-5	0.895 ^a^ ± 0.178	NQ	0.791 ^a^ ± 0.136	0.867 ^a^ ± 0.147	Waxy, green, orange, aldehydic, rose, mushroom
10	Undecanal	1639	112-44-7	0.062 ^a^ ± 0.014	NQ	NQ	NQ	Waxy, soapy, floral, aldehydic, citrus, green, fatty, fresh laundry
11	Ethyl 4-trans-decenoate	1729	76649-16-6	0.372 ^a^ ± 0.046	NQ	NQ	NQ	Green, fruity Waxy, cognac
12	Acetic acid, decyl ester	1744	112-17-4	0.036 ^a^ ± 0.005	NQ	NQ	0.057 ^b^ ± 0.009	Waxy, clean, fresh laundered cloths, citrus, soapy
13	(E)-methyl geranate	1759	1189-09-9	0.617 ^a^ ± 0.092	0.386 ^b^ ± 0.063	0.127 ^c^ ± 0.021	0.325 ^b^ ± 0.057	Waxy, green Fruity, flower
14	Alpha-terpineol	1762	10482-56-1	0.043 ^a^ ± 0.006	0.02 ^b^ ± 0.003	0.038 ^a^ ± 0.004	0.095 ^c^ ± 0.013	Lilac, floral, terpenic
15	Cyclooctane	1799	292-64-8	0.987 ^a^ ± 0.167	NQ	0.377 ^b^ ± 0.073	NQ	Camphor odor
16	Phenethyl isobutyrate	1956	103-48-0	0.057 ^a^ ± 0.011	NQ	NQ	NQ	Floral, fruity, rose, tea, peach, pastry
17	2-Acetylpyrrole	1978	1072-83-9	0.046 ^a^ ± 0.008	NQ	NQ	NQ	Musty, nut, skin, maraschino cherry, coumarinic, licorice, walnut, bready
18	Gamma-nonanolactone	2025	104-61-0	0.053 ^a^ ± 0.012	NQ	NQ	NQ	Coconut, creamy, waxy, sweet, buttery, oily
19	Ethyl acetate	891	141-78-6	1.638 ^a^ ± 0.347	1.761 ^a^ ± 0.331	2.472 ^b^ ± 0.421	2.375 ^b^ ± 0.397	Ethereal, fruity, sweet, weedy, green
20	Ethanol	895	64-17-5	14.774 ^b^ ± 2.358	12.073 ^a^ ± 1.872	18.836 ^d^ ± 2.468	13.821 ^ab^ ± 2.181	Strong alcoholic, ethereal, medical
21	Ethyl butyrate	1045	105-54-4	0.248 ^a^ ± 0.026	0.214 ^b^ ± 0.031	0.268 ^c^ ± 0.038	0.217 ^b^ ± 0.0025	Fruity, juicy, fruit, pineapple, cognac
22	Propan-1-ol	1059	71-23-8	0.080 ^a^ ± 0.012	0.065 ^b^ ± 0.011	0.083 ^a^ ± 0.012	0.090 ^c^ ± 0.021	Alcoholic, fermented fusel, musty
23	Ethyl isovalerate	1062	108-64-5	NQ	NQ	NQ	0.011 ^a^ ± 0.001	Fruity, sweet, apple, pineapple, tutti frutti
24	Isobutyl isobutyrate	1068	97-85-8	NQ	NQ	NQ	0.02 ^a^ ± 0.002	Ethereal, fruity, tropical fruit, pineapple, grape skin, banana
25	Isobutyl alcohol	1078	78-83-1	0.506 ^a^ ± 0.083	0.620 ^b^ ± 0.096	0.504 ^a^ ± 0.092	0.632 ^b^ ± 0.101	Ethereal, winey, cortex
26	Isoamyl acetate	1110	123-92-2	4.213 ^a^ ± 0.7235	6.493 ^b^ ± 0.925	10.603 ^c^ ± 1.532	12.447 ^cd^ ± 1.826	Sweet, fruity, banana, solvent
27	Myrcene	1122	123-35-3	0.025 ^a^ ± 0.003	0.019 ^b^ ± 0.002	0.029 ^c^ ± 0.003	0.048 ^d^ ± 0.005	Peppery, terpene, spicy, balsam, plastic
28	Isopentyl isobutyrate	1176	2050-01-3	0.018 ^a^ ± 0.002	NQ	0.123 ^b^ ± 0.025	0.166 ^c^ ± 0.037	Fruity, ethereal, tropical, green grape, cherry, unripe banana, apple, cocoa
29	3-Methyl-1-butanol	1200	123-51-3	13.623 ^a^ ± 1.793	11.607 ^b^ ± 1.467	13.146 ^a^ ± 1.983	11.17 ^b^ ± 1.249	Fusel, oil, alcoholic, whiskey, fruity, banana
30	2-Pentylfuran	1213	3777-69-3	NQ	NQ	NQ	0.01 ^a^ ± 0.001	Fruity, green, earthy, beany, vegetable, metallic
31	Ethyl caproate	1219	123-66-0	4.851 ^a^ ± 0.865	3.433 ^b^ ± 0.547	6.19 ^c^ ± 1.223	4.319 ^ab^ ± 0.792	Sweet, fruity, pineapple, waxy, green, banana
32	3,7-Dimethyl-1	1221	13877-91-3	NQ	NQ	NQ	0.018 ^a^ ± 0.003	Citrus, tropical, green, terpene, woody, green
33	Hexyl acetate	1228	142-92-7	0.079 ^a^ ± 0.013	0.135 ^b^ ± 0.028	0.149 ^bc^ ± 0.033	0.02 ^d^ ± 0.003	Fruity, green, apple, banana, sweet
34	2-Methylbutyl 2-methylbutyrate	1230	2445-78-5	NQ	NQ	NQ	0.024 ^a^ ± 0.003	Sweet, fruity, ester, berry, green, waxy, apple
35	Hexyl methyl ketone	1241	111-13-7	0.075 ^a^ ± 0.013	0.018 ^b^ ± 0.003	0.066 ^c^ ± 0.012	0.066 ^c^ ± 0.013	Earthy, weedy, natural, woody, herbal
36	Isopentyl isopentanoate	1273	659-70-1	NQ	NQ	NQ	0.035 ^a^ ± 0.005	Sweet, fruity, green, ripe, apple, jammy, tropical
	Ethyl heptanoate	1281	106-30-9	0.151 ^a^ ± 0.024	0.100 ^b^ ± 0.013	0.116 ^ba^ ± 0.015	0.274 ^c^ ± 0.044	Fruity, pineapple, cognac, rum wine
38	Hexanol	1299	111-27-3	0.059 ^a^ ± 0.009	0.039 ^b^ ± 0.006	0.037 ^b^ ± 0.005	NQ	Ethereal, fusel, oil, fruity, alcoholic, sweet, green
39	(Z)-4-Heptenal	1331	6728-31-0	NQ	NQ	0.027 ^a^ ± 0.004	0.035 ^b^ ± 0.006	Oily, fatty, green, dairy, milky, creamy
40	2-Hthylhexyl acetate	1369	103-09-3	0.021 ^a^ ± 0.003	0.020 ^a^ ± 0.003	NQ	NQ	Earthy, herbal, humus, undergrowth
41	Nonyl aldehyde	1391	124-19-6	NQ	NQ	0.073 ^a^ ± 0.012	0.197 ^b^ ± 0.037	Waxy, aldehydic, rose, fresh, orris, orange peel, fatty, peely
42	Ethyl caprylate	1430	106-32-1	19.674 ^a^ ± 3.85	19.791 ^a^ ± 4.03	17.016 ^ab^ ± 3.42	15.357 ^b^ ± 2.86	Fruity, wine, waxy, sweet, apricot, banana, brandy, pear
43	Furfural	1482	98-01-1	NQ	NQ	0.321 ^a^ ± 0.064	0.144 ^b^ ± 0.022	Sweet, woody, almond, fragrant, baked bread
44	Octyl acetate	1503	112-14-1	0.197 ^a^ ± 0.033	0.217 ^a^ ± 0.029	0.414 ^b^ ± 0.061	0.687 ^c^ ± 0.106	Green, earthy, mushroom, herbal, waxy
45	Camphor	1558	76-22-2	NQ	0.019 ^a^ ± 0.003	NQ	NQ	Camphoreous
46	2-Nonanol	1567	628-99-9	0.140 ^a^ ± 0.028	0.056 ^b^ ± 0.009	0.034 ^c^ ± 0.005	NQ	Waxy, green, creamy, citrus, orange, cheese, fruity
47	Ethyl nonanoate	1578	123-29-5	0.269 ^a^ ± 0.051	0.321 ^a^ ± 0.063	0.098 ^b^ ± 0.0156	0.054 ^c^ ± 0.008	Fruity, rose, waxy, rum, wine, natural, tropical
48	Linalool	1588	78-70-6	0.539 ^a^ ± 0.088	0.453 ^b^ ± 0.072	0.474 ^b^ ± 0.078	0.680 ^c^ ± 0.117	Citrus, floral, sweet, bois de rose, woody, green, blueberry
49	Octanoicacid	1592	5461-6-3	0.017 ^a^ ± 0.003	0.035 ^b^ ± 0.006	NQ	NQ	Fruity, green, oily, floral
50	Carbonochl	1633	7452-59-7	NQ	0.586 ^a^ ± 0.089	NQ	NQ	Sugar cane aroma, fruity
51	2-Decanol	1656	1120-06-5	0.030 ^a^ ± 0.005	0.017 ^b^ ± 0.002	NQ	0.051 ^c^ ± 0.009	NF
52	Ethyl caprate	1690	110-38-3	6.257 ^a^ ± 1.331	4.795 ^b^ ± 0.874	1.401 ^c^ ± 0.258	2.275 ^c^ ± 0.367	Sweet, waxy, fruity, apple, grape, oily, brandy
53	Acetophenone	1705	98-86-2	NQ	0.032 ^a^ ± 0.005	NQ	NQ	Sweet, pungent, hawthorn, mimosa, almond, acacia, chemical
54	Myrcene	1721	123-35-3	NQ	NQ	0.357 ^a^ ± 0.053	0.191 ^b^ ± 0.031	Peppery, terpene, spicy, balsam, plastic,
55	Ethyl benzoate	1739	93-89-0	0.082 ^a^ ± 0.012	0.091 ^b^ ± 0.014	0.056 ^c^ ± 0.008	0.064 ^d^ ± 0.009	Fruity, dry, musty, sweet, wintergreen
56	ethyl 9-decenoate	1748	67233-91-4	4.062 ^a^ ± 0.879	3.96 ^a^ ± 0.697	0.369 ^b^ ± 0.652	0.725 ^b^ ± 0.117	Fruity, fatty
57	2-Dodecanol	1771	10203-28-8	0.205 ^a^ ± 0.031	0.139 ^b^ ± 0.024	NQ	NQ	NF
58	4-Ethylbenzaldehyde	1782	4748-78-1	NQ	0.038 ^a^ ± 0.005	NQ	NQ	Bitter, almond, sweet, anise
59	Geranyl acetate	1790	105-87-3	0.031 ^a^ ± 0.005	0.029 ^a^ ± 0.004	0.034 ^a^ ± 0.005	0.030 ^a^ ± 0.006	Floral, rose, lavender, green, waxy
60	Decyl alcohol	1800	112-30-1	NQ	0.678 ^a^ ± 0.098	NQ	0.488 ^b^ ± 0.078	Fatty, waxy, floral, orange, sweet, clean, watery
61	Citronellol	1804	106-22-9	0.508 ^a^ ± 0.089	0.346 ^b^ ± 0.064	0.352 ^b^ ± 0.059	0.174 ^c^ ± 0.031	Floral, leather, waxy, rose bud, citrus
62	Ethyl phenylacetate	1807	101-97-3	0.027 ^a^ ± 0.004	0.049 ^b^ ± 0.001	0.029 ^a^ ± 0.005	0.036 ^c^ ± 0.004	Sweet, floral, honey, rose, balsam, cocoa
63	Nerol	1810	106-25-2	0.028 ^a^ ± 0.004	0.032 ^ab^ ± 0.006	NQ	0.009 ^c^ ± 0.001	Sweet, natural, neroli, citrus, magnolia
64	Beta-damascenone	1816	23726-93-4	0.087 ^a^ ± 0.012	NQ	0.275 ^b^ ± 0.040	0.333 ^c^ ± 0.058	Apple, rose, honey, tobacco, sweet
65	Ethyl laurate	1819	106-33-2	0.159 ^a^ ± 0.027	0.219 ^b^ ± 0.031	0.019 ^c^ ± 0.003	NQ	Sweet, waxy, floral, soapy, clean
66	Geraniol	1821	106-24-1	0.103 ^a^ ± 0.027	0.054 ^b^ ± 0.009	0.066 ^c^ ± 0.010	0.095 ^d^ ± 0.012	Sweet, floral, fruity, rose, waxy, citrus
67	Geranylacetone	1825	689-67-8	0.170 ^a^ ± 0.030	0.068 ^b^ ± 0.011	0.075 ^b^ ± 0.013	NQ	Fresh, rose, leaf, floral, green, magnolia, aldehydic, fruity
68	Trimethyl pentanyl diisobutyrate	1828	6846-50-0	NQ	0.077 ^a^ ± 0.013	0.039 ^b^ ± 0.006	NQ	NF
69	Ethyl hydrocinnamate	1930	2021-28-5	0.070 ^a^ ± 0.010	0.108 ^b^ ± 0.017	0.072 ^a^ ± 0.013	0.068 ^a^ ± 0.013	Hyacinth, rose, honey, fruity, rum
70	Butylated hydroxytoluene	1936	128-37-0	NQ	0.013 ^a^ ± 0.002	0.486 ^b^ ± 0.071	NQ	Mild, phenolic, camphor
71	DMS	767	75-18-3	0.019 ^ba^ ± 0.003	0.085 ^c^ ± 0.011	0.010 ^a^ ± 0.001	0.025 ^b^ ± 0.004	Sulfury, onion, sweet, corn, vegetable, cabbage, tomato, green, radish
72	Heptyl acetate	1356	112-06-1	0.066 ^a^ ± 0.012	0.113 ^b^ ± 0.017	0.183 ^c^ ± 0.030	0.134 ^bc^ ± 0.023	Fresh, green, rum, ripe, fruit, pear, apricot, woody
73	Benzyl alcohol	1571	100-51-6	0.033 ^a^ ± 0.005	0.077 ^b^ ± 0.012	0.117 ^c^ ± 0.020	NQ	Floral, rose, phenolic, balsamic
74	Isoamyl octanoate	1725	2035-99-6	0.101 ^a^ ± 0.017	0.302 ^b^ ± 0.048	NQ	0.015 ^c^ ± 0.002	Sweet, oily, fruity, green, soapy, pineapple, coconut
75	Phenethyl acetate	1813	103-45-7	2.611 ^a^ ± 0.415	8.201 ^cb^ ± 1.311	6.086 ^b^ ± 0.985	9.984 ^c^ ± 1.837	Floral, rose, sweet, honey, fruity, tropical,
76	Phenethyl alcohol	1941	60-12-8	9.061 ^a^ ± 1.364	12.49 ^b^ ± 2.074	9.913 ^a^ ± 1.693	10.503 ^ab^ ± 1.795	Floral rose, dried rose, flower, rose water
77	2,4-Di-t-butylphenol	2341	96-76-4	0.127 ^a^ ± 0.016	0.357 ^b^ ± 0.063	0.125 ^a^ ± 0.020	0.105 ^c^ ± 0.018	Phenolic

Note: Relative content data in this table are expressed as mean (*n* = 5) ± standard deviation, using Duncan’s test, and differences in superscript letters are significant (*p* < 0.05), NQ means that the substance is not detectable.

**Table 10 foods-13-02339-t010:** Comparison of major aged aldehydes in five kinds of lager after 9 days of aging storage at 45 °C.

Aging-Related Substance Content (µg/L)						
	2-Furaldehyde	Acetaldehyde	Isobutyraldehyde	Valeraldehyde	Phenylacetaldehyde	Total Aldehyde
Unaged commercially available lager (CAL1)	NQ	3.04 ± 0.413	38.385 ± 3.125	5.386 ± 0.281	6.133 ± 0.818	52.944 ^b^ ± 4.696
Aged commercially available lagers (CAL_1_)	66.887 ± 8.423	75.414 ± 9.052	49.914 ± 5.747	17.257 ± 1.425	8.996 ± 0.755	218.468 ^a^ ± 25.402
Unaged commercially available lager (CAL_2_)	NQ	7.08 ± 0.629	15.394 ± 1.724	16.841 ± 2.218	4.823 ± 0.293	44.138 ^b^ ± 4.864
Unaged commercially available lager (CAL_2_)	48.792 ± 5.466	92.348 ± 10.133	38.266 ± 2.635	25.978 ± 3.935	12.759 ± 0.828	218.143 ^a^ ± 22.997
Unaged LS	NQ	28.592 ± 3.981	2.7126 ± 0.416	14.197 ± 1.325	1.861 ± 0.252	47.366 ^b^ ± 5.78
Aged LS	11.412 ± 2.793	77.862 ± 9.051	8.758 ± 1.520	13.886 ± 2.047	10.744 ± 0.973	122.662 ^c^ ± 16.384
Unaged OWBL	NQ	69.306 ± 7.248	7.872 ± 0.923	9.162 ± 1.325	3.743 ± 0.287	90.083 ^d^ ± 11.125
Aged OWBL	22.503 ± 2.83	139.78 ± 18.723	14.657 ± 1.937	16.035 ± 1.692	9.622 ± 0.983	202.597 ^a^ ± 26.165

Note: Total Aldehyde data in this table are expressed as mean (*n* = 5) ± standard deviation, using Duncan’s test, and differences in superscript letters are significant (*p* < 0.05), NQ means that the substance is not detectable.

## Data Availability

The original contributions presented in the study are included in the article/Appendix A, further inquiries can be directed to the corresponding author.

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
