# Peer review of "A Preliminary Study on the Effect of Adding Sugarcane Syrup on the Flavor of Barley Lager Fermentation"

_foods, 2024, doi:10.3390/foods13152339_

Round 1

Reviewer 1 Report (Previous Reviewer 1)

Comments and Suggestions for Authors

The resubmitted manuscript has been expanded compared to the original article and has been supplemented with missing information, especially with regard to methodology.

But I have the following comments:

The abbreviations should be explained in the abstract - LS, OWBL. Last sentence - what is meant by "instead of partial wort"?

Again, however, I recommend that the authors reconsider the text of the conclusion, which is too long, because the conclusion should answer the aim of the present study, i.e. briefly summarise what new findings were found and how the findings advance our knowledge in the field, and not repeat the results obtained.

I also recommend that the authors consider presenting the higher amino acid utilisation with the addition of cane syrup in the abstract and conclusion. Higher amino acid utilisation with the addition of any sugar supplement is well known.

As in the original manuscript, I recommend unifying terminology and using commonly known terms - colour (not chromaticity), wort extract, higher alcohols (not "senior alcohols"), etc.

In the introduction and the following text, "carbon-based substances" are named as active substances in sensory ageing. Are carbonyls meant? Because carbon based are, for example, sugars.

Author Response

The abbreviations should be explained in the abstract - LS, OWBL. Last sentence - what is meant by "instead of partial wort"? ——In this paper, part of the cane sugar juice was used for beer fermentation instead of part of the wort, not additional cane syrup was added for beer fermentation, and all the relative data is the same, the wort concentration can also be called the brix, and in this paper, the wort concentration is the brix measured by mixing the cane sugar juice and the wort according to the proportions. The abstract section has been labeled.

Again, however, I recommend that the authors reconsider the text of the conclusion, which is too long, because the conclusion should answer the aim of the present study, i.e. briefly summarise what new findings were found and how the findings advance our knowledge in the field, and not repeat the results obtained.——The text of the conclusions has been redacted.

I also recommend that the authors consider presenting the higher amino acid utilisation with the addition of cane syrup in the abstract and conclusion. Higher amino acid utilisation with the addition of any sugar supplement is well known.——This you are right, it does enhance the utilization of free amino acids in lagers if you add sugar classified substances directly, but cane syrup is not an ordinary beer adjunct, it is rich in free amino acids, and because of the abundance of amino acids carried in the ingredient itself, and the fermentation that imparts the different flavor substances in the beer, I still think these are necessary.

As in the original manuscript, I recommend unifying terminology and using commonly known terms - colour (not chromaticity), wort extract, higher alcohols (not "senior alcohols"), etc.——It has been appropriately revised as per your suggestions.

In the introduction and the following text, "carbon-based substances" are named as active substances in sensory ageing. Are carbonyls meant? Because carbon based are, for example, sugars.——I'm sorry I misspelled it, it should be carbonyl compounds, which are aldehydes and ketones that can be quantified as aging substances.

Reviewer 2 Report (New Reviewer)

Comments and Suggestions for Authors

In this paper, the authors have presented a preliminary study on the effects of brewing lager beer with sugar cane syrup instead of some malt and on the influence of this syrup on the quality of the resulting beer. The title and abstract actually reflect the content of the paper. The paper contains a large amount of information, but is riddled with many grammatical errors, making the paper itself quite difficult to read.

Overall, this is an interesting manuscript, it need take the following remarks into account:

-      the sentence in lines 52-53 is repeated

-    since the authors later mention the aroma of sugar cane syrup, it might be good to mention in the introduction the volatile aroma compounds identified in sugar cane and sugar cane syrup so that it is easier to follow the results obtained later on

-   parts 232 to 254 should be placed before Table 2, as they refer to and explain Table 1.

-   in the entire results chapter, the authors should state exactly which table the results refer to (table number)

-  in line 252 is written …”I was….”  -  this is about a group of authors, why is it written in the singular?

-   Text in line 281-284 already written in line 214 and 215

-    line 291: “was between 0.1 and 0.1 mg/L”

-    line 331-332: reversed values for fructose and glucose, according to table 7.

-    line 341: “…yeast toAt the end…”..  - what does that mean?

-    line 396: Figure 1 instead Figure 5

-    line 402: “…indicating This indicates that…”

-    line 409: ..W1W and W1W… - repeated

-    line 413-414 :  The  amount of SO2  produced depends on the yeast strain, wort concentration and fermentation process are unnecessary. It is the same sentence as the previous one.

-   line 419-422 repeated sentences

-   line 432-434 repeated sentences

-   line 470 Table 10 instead Table 6

The "References" chapter is not completely written according to the journal's instructions for authors, and more than half of the cited references are not current (mostly in the last 5 years).

Comments on the Quality of English Language

The language and phrasing are unclear and a bit confusing, so it is necessary moderate English language.

Author Response

Question:

  1.   the sentence in lines 52-53 is repeated ——Corrected.
  2.  since the authors later mention the aroma of sugar cane syrup, it might be good to mention in the introduction the volatile aroma compounds identified in sugar cane and sugar cane syrup so that it is easier to follow the results obtained later on——A small amount of literature on sugarcane flavor has been added to the foreword, and in fact relatively few articles have explored sugarcane and molasses flavors..
  3. parts 232 to 254 should be placed before Table 2, as they refer to and explain Table 1.——WE have put the sentence in its proper place..
  4. in the entire results chapter, the authors should state exactly which table the results refer to (table number)——Table headers and table contents have been checked.
  5. in line 252 is written …”I was….”  -  this is about a group of authors, why is it written in the singular?——It has been modified.
  6. Text in line 281-284 already written in line 214 and 215——Duplicate pages have been removed.
  7. line 291: “was between 0.1 and 0.1 mg/L——Corrected.
  8. line 331-332: reversed values for fructose and glucose, according to table 7.——Corrected .
  9. line 341: “…yeasttoAt the end…”..  - what does that mean?——The entire paragraph has been rewritten from the ground up.
  10. line 402: “…indicating This indicates that——Corrected.
  11. line 409: ..W1W and W1W… - repeated——Corrected.
  12. line 413-414 :  The  amount of SOproduced depends on the yeast strain, wort concentration and fermentation process are unnecessary. It is the same sentence as the previous one.——WE have put the sentence in its proper place.
  13. line 419-422 repeated sentences——WE have put the sentence in its proper place.
  14. line 432-434 repeated sentences——WE have put the sentence in its proper place.
  15. line 470 Table 10 instead Table 6——corrected

 The "References" chapter is not completely written according to the journal's instructions for authors, and more than half of the cited references are not current (mostly in the last 5 years).——I'm sorry to say that all of these articles are searchable on web ofscience, and these citations include

Round 2

Reviewer 2 Report (New Reviewer)

Comments and Suggestions for Authors

The authors have accepted most of the corrections, although there are still some technical and grammatical errors that I hope will be corrected, as the authors themselves stated in the statement of purpose. As for the literature, I still maintain that it is not completely written according to the instructions for the authors of the journal itself (for example, the year must be in bold, volumes in italics...), but I think that there are small corrections that the authors can fix in the final version of his work.

Comments on the Quality of English Language

Some further corrections are needed to improve the English language.

This manuscript is a resubmission of an earlier submission. The following is a list of the peer review reports and author responses from that submission.

Round 1

Reviewer 1 Report

Comments and Suggestions for Authors

My comment was about the too brief description of the preparation of experimental beers, what is important for reproducibility and further interpretation of the experimental results. The authors just moved the text without significant changes to the supplement (and I couldn't find a reference to the supplement in the methodology). In the actual description of the experiment, a number of important pieces of information are missing, wort boiling operation (hopping), volume of the experiment (fermenter) etc. Some of the wording in the description is ambiguous or repetitive. The description of the preparation of the experimental beers should therefore be reconsidered.

L18 - The abstract has been reformulated, the formulation “… to ordinary whole wheat beer, sugarcane beer …”suggests a beer made from wheat (Triticum aestivum).

L 555 – “Table 12. Comparison of major aged aldehydes in five kinds of lager after 9 days of aging storage at 45℃.” The table shows the results of only three beers and it is also not clear which commercial beer is listed.

The comprehensive conclusion has been significantly expanded from the original version. I recommend reconsidering the text, because the conclusion should answer the aim of the present study, so conclude what new findings were found and how the findings advance our knowledge in the field, and do no reiterate the results obtained.

 Author Response

Thank you for giving us the opportunity to submit a revised draft of the manuscript “A preliminary study on the effect of adding sugarcane syrup on the flavor of barley lager fermentation” for publication in the Foods.  We have studied comments carefully and have made correction which we hope meet with approval.

We have incorporated most of the suggestions made by the reviewers. We will mark your comments and give you a solution.

Response to comment: 

Question :

  1. In the actual description of the experiment, a number of important pieces of information are missing, wort boiling operation (hopping), volume of the experiment (fermenter) etc. Some of the wording in the description is ambiguous or repetitive. The description of the preparation of the experimental beers should therefore be reconsidered.——The specific method of beer fermentation has been written in the supplementary.
  2. L18 - The abstract has been reformulated, the formulation “… to ordinary whole wheat beer, sugarcane beer …”suggests a beer made from wheat (Triticum aestivum).——corrected.
  3. L 555 – “Table 12. Comparison of major aged aldehydes in five kinds of lager after 9 days of aging storage at 45℃.” The table shows the results of only three beers and it is also not clear which commercial beer is listed.——It has been changed to a more detailed table.
  4. The comprehensive conclusion has been significantly expanded from the original version. I recommend reconsidering the text, because the conclusion should answer the aim of the present study, so conclude what new findings were found and how the findings advance our knowledge in the field, and do no reiterate the results obtained.——The conclusion has been modified.

We will be completing the revisions to the article and submitting it again to Foods shortly, and we hope that this time we can fulfill your request. Again, we sincerely appreciate your advice and instruction.

Yours sincerely

Hechao Lv

Guangxi University, School of Light Industry and Food Engineering

Reviewer 2 Report

Comments and Suggestions for Authors

The research article it contains quite a few errors, spelling mistakes:

1.      Keywords:  sugarcane syrup  beer  process optimization   Volatile profifiles  aroma stability

2.      do not use colons for titles and sub-titles example, Sensory evaluation experiment:

3.      take care of the punctuation marks, the space between these, example: Detection Methods :

The mixed standards were four alcohols:n-propanol

capillary column (60 m0.25 125 mm0.25 m)

comparison with the wavelength of 517nm

2 g NaCL

A dilute solution of PFBOA was prepared by mixing 100 L (6 g/L)

the optimized conditions are andA2B2C2D2

one-way experiment.A2B2C2D

is considered feasible[4,23]

4.      in some cases the measurement units are missing, or it is written without space between the number and the unit, example: (4.6×250)

5.      The Gc condition is not written correct, “GC conditions: the column was an Agilent DB-Wax capillary column (60 m0.25 125 mm0.25 m); carrier gas: helium, 99.999%; constant flow mode: 1.0 mL/min; etc.: the initial temperature was 4 min at 35°C, and the initial temperature was 4 min at 3°C/min”, the column heating profile is missing from the description.

6.      The initial letter from the subtitle should be capital in each case, example: 3.1. influence of process parameters on the finished beer

3.3.2. free amino nitrogen:

7.      If the authors mention some comment in the table, shortened with letters, it should be explained at the end of the table what they wanted to express. Examples table 1, 5.6d±0.3, what the letter d explains?

8.      Table 6: physical and chemical requirements of pale lager. There are some mistakes: space in between the project names, examples: CO2(Mass fraction)

9.      Table 7 the same problem, example: 10 °P Ordinary all-barley beer(mg/100 g)

Table 7. Utilization of amino nitrogen in sugarcane beer and common beer.

10.   There are lot of phrases which are repeat the idea, example:

GC-MS experiments are needed to check the specific flavor substances in the lager. The GC-MS test is needed to examine the specific flavor 462 components and proportions in the lager.

11.   Table 8, Table 9 and Table 10, the amount of compounds find in samples are expressed in? Relative %? (Area %?)

12.   In Table 9 some of the compounds name are written partially or incorrect, example: 3,7-Dimethyl-1…?, 2-Hthylhexyl acetate, Carbonochl, Octanoicacid, 

Author Response

Response to Reviewers

Dear Reviewer

Thank you for giving us the opportunity to submit a revised draft of the manuscript “A preliminary study on the effect of adding sugarcane syrup on the flavor of barley lager fermentation” for publication in the Foods. We appreciate the time and effort that you and the reviewers dedicated to providing feedback on our manuscript and are grateful for the insightful comments on and valuable improvements to our paper.

We have incorporated most of the suggestions made by the reviewers. We will mark your comments and give you a solution.

Response to comment: 

Question:

  1. Keywords:  sugarcane syrup  beer  process optimization   Volatile profifiles  aroma stability ——
  2. do not use colons for titles and sub-titles example, Sensory evaluation experiment:——
  3. take care of the punctuation marks, the space between these, example: Detection Methods :.——
  4. The mixed standards were four alcohols: n-propanol——
  5. capillary column (60 m0.25 125 mm0.25 m)——
  6. comparison with the wavelength of 517nm——
  7. 2 g NaCL——
  8. A dilute solution of PFBOA was prepared by mixing 100 L (6 g/L)——Corrected .
  9. the optimized conditions are andA2B2C2D2——
  10. one-way experiment.A2B2C2D——
  11. is considered feasible[4,23] ——
  12. The Gc condition is not written correct, “GC conditions: the column was an Agilent DB-Wax capillary column (60 m0.25 125 mm0.25 m); carrier gas: helium, 99.999%; constant flow mode: 1.0 mL/min; etc.: the initial temperature was 4 min at 35°C, and the initial temperature was 4 min at 3°C/min”, the column heating profile is missing from the description.——It has been rewritten to the correct statement.
  13. The initial letter from the subtitle should be capital in each case, example: 3.1. influence of process parameters on the finished beer——
  14. 3.2. free amino nitrogen:——corrected .
  15. If the authors mention some comment in the table, shortened with letters, it should be explained at the end of the table what they wanted to express. Examples table 1, 5.6d±0.3, what the letter d explains?——Each letter in the table represents a different significance, with different letters representing significant differences between different data, while the advanced alcohol column is not required for significance comparisons, but merely to support the data for subsequent GC-MS.
  16. Table 6: physical and chemical requirements of pale lager. There are some mistakes: space in between the project names, examples: CO2(Mass fraction)——corrected
  17. Table 7 the same problem, example: 10 °P Ordinary all-barley beer(mg/100 g)——
  18. Table 7. Utilization of amino nitrogen in sugarcane beer and common beer.——
  19. GC-MS experiments are needed to check the specific flavor substances in the lager. The GC-MS test is needed to examine the specific flavor 462 components and proportions in the lager.——More than content has been deleted.
  20. Table 8, Table 9 and Table 10, the amount of compounds find in samples are expressed in? Relative %? (Area %?)——The results of a gas chromatogram usually indicate the percentage of each compound in the sample. This percentage can be used to determine the relative amounts of individual compounds.
  21. In Table 9 some of the compounds name are written partially or incorrect, example: 3,7-Dimethyl-1…?, 2-Hthylhexyl acetate, Carbonochl, Octanoicacid, ——Each compound may be spelled with a number of proper nouns, and I just chose one of them.

We will be completing the revisions to the article and submitting it again to Foods shortly, and we hope that this time we can fulfill your request. Again, we sincerely appreciate your advice and instruction.

Yours sincerely

Hechao Lv

Guangxi University, School of Light Industry and Food Engineering

Reviewer 3 Report

Comments and Suggestions for Authors

Please correct senior alcohol in "higher alcohols" throughout the text.

Please revise the English.

Comments on the Quality of English Language

English language should be improved.

Author Response

Manuscript foods-2857176

Response to Reviewers

Dear Reviewer

Thank you for giving us the opportunity to submit a revised draft of the manuscript “A preliminary study on the effect of adding sugarcane syrup on the flavor of barley lager fermentation” for publication in the Foods. We appreciate the time and effort that you and the reviewers dedicated to providing feedback on our manuscript and are grateful for the insightful comments on and valuable improvements to our paper.

We have incorporated most of the suggestions made by the reviewers. We will mark your comments and give you a solution.

Response to comment: 

Question:

  1. the abstract does not give a background and novelty of the work and .  ——Abstracts have been modified.
  2. lthe methodology is still not scientifically written——Method details have been modified.
  3. the pdf file has no appendix——The fermentation process of beer was placed in the Supplementary .

We will be completing the revisions to the article and submitting it again to Foods shortly, and we hope that this time we can fulfill your request. Again, we sincerely appreciate your advice and instruction.

Yours sincerely

Hechao Lv

Guangxi University, School of Light Industry and Food Engineering

Reviewer 4 Report

Comments and Suggestions for Authors

The manuscript evaluated the impact of sugar cane syrup as an additive in lager beer properties. The topic is relevant and within the scope of the journal. However, the revised manuscript is still not well written. The authors did not follow all the requested corrections and did not send the changes highlighted in the manuscript, which made its revision very difficult. For example, the abstract does not give a background and novelty of the work and the methodology is still not scientifically written. Both corrections were asked in the first round of review. The authors mention an appendix in their reply letter, but the pdf file has no appendix. For these reasons I still consider that the manuscript is not in a scientific standard for publication.

Author Response

Manuscript foods-276186

Response to Reviewers

Dear Reviewer

Thank you for giving us the opportunity to submit a revised draft of the manuscript “A preliminary study on the effect of adding sugarcane syrup on the flavor of barley lager fermentation” for publication in the Foods. We appreciate the time and effort that you and the reviewers dedicated to providing feedback on our manuscript and are grateful for the insightful comments on and valuable improvements to our paper.

We have incorporated most of the suggestions made by the reviewers. We will mark your comments and give you a solution.

Response to comment: 

Question:

  1. the abstract does not give a background and novelty of the work and .  ——Abstracts have been modified.
  2. lthe methodology is still not scientifically written——Method details have been modified.
  3. the pdf file has no appendix——The fermentation process of beer was placed in the Supplementary .

We will be completing the revisions to the article and submitting it again to Foods shortly, and we hope that this time we can fulfill your request. Again, we sincerely appreciate your advice and instruction.

Yours sincerely

Hechao Lv

Guangxi University, School of Light Industry and Food Engineering

Round 2

Reviewer 4 Report

Comments and Suggestions for Authors

Authors do not provided a detailed letter containing the point by point answers to my comments of the first round, I just received the second round. The authors did not follow all the requested corrections and did not send the changes highlighted in the manuscript, which made its revision very difficult. However it is clear that the abstract still is the first version submmited and it does not give a background and novelty of the work and the methodology is still not scientifically written. Results does not have deviation bars. Both corrections were asked in the first round of review. For these reasons I still consider that the manuscript is not in a scientific standard for publication.

Author Response

Manuscript foods-276186

Response to Reviewer

Dear Reviewer:

Thank you for giving us the opportunity to submit a revised draft of the manuscript “A preliminary study on the effect of adding sugarcane syrup on the flavor of barley lager fermentation” for publication in the Foods.  We have studied comments carefully and have made correction which we hope meet with approval.

I apologize for not responding more specifically to your first review, and I will give you more specific solutions for your first and second reviews this time around.

Response to comment: 

Question :

  1. Regarding the absence of deviation bars in the results (e.g., Figures 1 ,2,and3)—— 1 and Fig. 2 only express the trend of fermentable sugars during the fermentation process, from Fig. 1, it can be seen that different concentrations of wort do not affect the proportion of fermentable sugars, but relatively, after the addition of cane juice in different proportions, some changes in the fermentable sugars in the fermentation matrix, Fig. 2 shows the amount of changes in the amount of fermentable sugars due to the addition of different fermentation matrices, perhaps not a great effect on the amount of sugar consumed, but the addition of cane syrup on the flavor of the lager beer mainly exists in the amino acid composition of the difference between the effects of the flavor of the beer.The amount of change of the different amino nitrogen species in it is expressed in Fig. 3, corresponding to the data in Table 7, where the trend of amino nitrogen change in different fermentation matrices was determined, and after narrowing down the graphs presented for the first time, it is more clearly seen that the addition of sugarcane molasses resulted in the yeast utilizing the free amino nitrogen higher than that of normal beer, and the different amino acids being utilized in the different periods of the fermentation are shown relatively intuitively in Fig. 3.
  2. The abstract needs to give the background of the work and the novelty of the study.——The abstract has been revised to highlight the context of the study and the main research objectives.
  3. Methodology needs to be rewritten in the language of science.——References were added to the methodology and statements were modified.

We will be completing the revisions to the article and submitting it again to Foods , and we hope that this time we can fulfill your request. Again, we sincerely appreciate your advice and instruction.

Yours sincerely

Hechao Lv

Guangxi University, School of Light Industry and Food Engineering
